# Stratospheric gravity waves in three high-resolution models and AIRS satellite observations

Phoebe Noble<sup>1</sup>, Haruka Okui<sup>1,2</sup>, Joan Alexander<sup>3</sup>, Manfred Ern<sup>4</sup>, Neil P. Hindley<sup>1</sup>, Lars Hoffmann<sup>5</sup>, Laura Holt<sup>3</sup>, Annelize van Niekerk<sup>6</sup>, Riwal Plougonven<sup>7</sup>, Inna Polichtchouk<sup>6</sup>, Claudia C. Stephan<sup>8</sup>, Martina Bramberger<sup>9</sup>, Milena Corcos<sup>3</sup>, William Putnam<sup>10</sup>, Christopher Kruse<sup>11</sup>, and Corwin J. Wright<sup>1</sup>

**Correspondence:** Phoebe Noble (pn399@bath.ac.uk)

**Abstract.** Advances in computational power and model development have enabled the generation of global high-resolution models. These new models can resolve a large proportion of gravity waves (GWs) explicitly, reducing reliance on subgrid parametrizations. GWs are vital components of the middle and upper atmosphere, they transport energy and momentum both horizontal and vertically, driving the atmospheric circulation. Evaluating the realism of these resolved waves is a crucial step in advancing future model development.

Here we provide the first global multi-model GW observational comparison that accounts for the observational filter. We assess the representation of stratospheric GWs in three high-resolution (3-5 km horizontal resolution) global free-running simulations (ICON, IFS and GEOS), for the period 20th January- 29th February 2020, against AIRS satellite observations.

Wave amplitudes are systematically lower in the models than observations, consistent with previous studies. GW occurrence rates are higher in all models than the observations, dominated by low amplitude waves in the models. During the first 10 days spatial patterns of GW occurrence rate, amplitudes and momentum flux agree across the models and observations but subsequently they diverge. Agreement is more consistent in the northern hemisphere (where orographic waves dominate) than in the southern hemispheric convective regions.

These results benchmark the current state of high-resolution modelling and demonstrate that whilst there are strengths in models' ability to capture the morphology of GWs (particularly orographically generated waves), there is room for improvement in modelling amplitudes, occurrence rates and zonal-mean flux magnitudes globally, with the largest discrepancies in the tropical convective regions.

<sup>&</sup>lt;sup>1</sup>University of Bath, Bath, UK

<sup>&</sup>lt;sup>2</sup>Department of Earth and Planetary Science, the University of Tokyo, Tokyo, Japan

<sup>&</sup>lt;sup>3</sup>Northwest Research Associates, Boulder, CO, USA

<sup>&</sup>lt;sup>4</sup>Institute of Climate and Energy Systems: Stratosphere (ICE-4), Forschungszentrum Jülich, Jülich, Germany

<sup>&</sup>lt;sup>5</sup>Jülich Supercomputing Centre, Forschungszentrum Jülich, Jülich, Germany

<sup>&</sup>lt;sup>6</sup>European Centre for Medium-Range Weather Forecasts, Reading, UK

<sup>&</sup>lt;sup>7</sup>École Polytechnique, Paris, France

<sup>&</sup>lt;sup>8</sup>Leibniz Institute of Atmospheric Physics at the University of Rostock, Kühlungsborn

<sup>&</sup>lt;sup>9</sup>NSF National Centre for Atmospheric Research

<sup>&</sup>lt;sup>10</sup>NASA, US

<sup>&</sup>lt;sup>11</sup>Research Aviation Facility, Earth Observing Laboratory, NSF National Center for Atmospheric Research

# 1 Introduction

Atmospheric gravity waves (hereafter referred to as GWs) are fundamentally important to middle and upper atmosphere dynamics. These waves, with horizontal wavelengths ranging from tens to thousands of kilometers, play a significant role in the vertical and horizontal coupling of the atmosphere (Alexander et al., 2010). They transport energy and momentum from their sources to where they break and are a key driver of the middle atmospheric circulation (Andrews et al., 1987). GWs are often much smaller than the grid-size of global general circulation models and so their effects are parametrized in such models. This means that due to computational limitations, instead of the model fully resolving the physics of the waves the effects of GWs are simplified. These parametrizations often include major assumptions, including that GWs only propagate directly vertically and that they have tropospheric sources only with no secondary generation of waves (Alexander et al., 2010; Plougonven et al., 2020).

Modern supercomputing capabilities have brought the ability to resolve the full GW spectrum potentially within reach. These models have resolutions of a few kilometers in the horizontal and of order 1 km in the vertical at stratospheric heights. This means that the reliance on GW parametrizations and their limiting assumptions will be greatly reduced. It is thus important to assess the realism of these resolved waves to consolidate model developments and test these assumptions.

Accordingly, in this study we compare stratospheric GWs simulated by three high-resolution models with observations from the nadir-viewing AIRS (Atmospheric Infrared Sounder) satellite-based instrument. We compare GWs globally for the boreal winter period from 20th January to 29th February 2020. Our model data comes from the DYAMOND (DYnamics of the Atmospheric general circulation Modeled On Non-hydrostatic Domains) high-resolution simulations (Stevens et al., 2019a). These runs explicitly do not use convective parametrizations and should in principle be able to accurately simulate the majority of GWs with minimal reliance on parametrization.

A recent review by Achatz et al. (2024), highlights both the importance and challenges of evaluating GW permitting simulations against observations. In observations, we only have estimates of gravity wave momentum flux (GWMF) derived temperatures and cannot see the full spectrum of waves (the observational filter - Alexander (1998)). Achatz et al. (2024) recommend that 'measurement' methods, wherein the model atmosphere is sampled as though viewed by the observing instrument, can help get round these limitations and provide a fair comparison in model-observational comparisons. This is the approach that we take in this study, we sample the model atmosphere as if viewed by the AIRS satellite (Wright and Hindley, 2018). This allows us to take into account the observational filter of AIRS when assessing the representation of GWs in the models. Our work provides the first global multi-model GW observational comparison which takes into account the observational filter.

Satellites provide the ideal observational platform for evaluating model performance across large spatial scales. Notable global GW satellite to model comparisons include Geller et al. (2013), Preusse et al. (2014) and Gong et al. (2015). However, since these three studies there have been significant model and computational advancements, notably, a drive towards higher horizontal resolutions. More recently, Stephan et al. (2019) carried out a detailed inter-comparison of six global-convection permitting simulations for the DYAMOND summer model project for the period August 2016 and compared to the GRACILE infrared limb sounder GW climatology (Ern et al., 2018) and AIRS observations. Their results show that whilst many large-

scale patterns of GW activity were represented well, there were significant differences especially in the magnitudes of GWMF. A crucial difference from this study is that we explicitly sample the model as though viewed by AIRS before our comparison to account for the observational filter. This was the approach taken in Okui et al. (2023) where they carried out a single model comparison to AIRS observations.

On a regional scale, model resolution is not so much of a limiting factor. The AIRS instrument, with exceptionally high horizontal resolution provides detailed observations to test these new high-resolution models against. Model-AIRS studies include Hindley et al. (2021) (South Georgia, Southern Ocean), Grimsdell et al. (2010) (Australia), Lear et al. (2024) (Asia) and Kruse et al. (2022) (the Andes, South America). We discuss all these global and regional model-observational comparison studies in further detail later in the discussion section of this work.

Another key limitation of previous model-observational comparison studies is the presence of noise in observations. In this work, we apply a new machine learning (ML) algorithm developed by Okui et al. (2025) to identify GWs in both the original AIRS observations and the model-sampled data. This new wave identification allows for the direct comparison of identified GW events instead of bulk averages which would include times where no waves are observed.

This study assesses the representation of GWs in three high-resolution models in boreal winter, specifically; 1) a 4-km run of the Integrated Forecast System (IFS) produced by the European Centre for Medium-range Weather Forecasts (ECMWF), hereafter known as IFS, 2) a 3-km run of the Goddard Earth Observing System model (GEOS) produced by NASA and hereafter referred to as GEOS, and 3), a 5-km run of the Icosahedral Non hydrostatic model, ICON.

In summary, we first sample the models as though viewed by the AIRS satellite instrument to take into account the observational filter (Wright and Hindley, 2018). Next, we apply three-dimensional spectral analysis to derive GW properties(Hindley et al., 2019; Wright et al., 2021). We then apply the machine learning detection method of Okui et al. (2025). This allows us to examine the GW occurrence rate and GW amplitudes in the AIRS observations and models globally. Additionally, we derive momentum fluxes and the distribution of horizontal and vertical wavenumbers of the GWs found in each dataset. Finally, we provide context by investigating in detail the background stratospheric wind and precipitation data across the 40-day period covered by the DYAMOND simulations. This work provides a detailed benchmark of the current state-of-the-art in high-resolution modelling as compared to global stratospheric observations provided by AIRS.

Our work is structured as follows. We first introduce the datasets used in Section 2. Section 3 describes the methods: model sampling, GW detection and derivation of GW properties. Section 4 provides the results and in Section 5 we discuss the implications of the results. Finally, we summarise and conclude the work in Section 6.

# 80 **2 Data**

# **2.1 AIRS**

We use observations from the satellite-based AIRS (Atmospheric InfraRed Sounder) instrument. AIRS is an instrument on-board the National Aeronautics and Space Administration's (NASA) Aqua satellite (Aumann et al., 2003; Chahine et al., 2006). Aqua was launched in 2002, and is part of NASA's 'A-train' satellite constellation orbiting the earth every  $\sim 100$  minutes in a

near-polar sun-synchronous orbit with fixed local time of  $\sim$ 01:30 (descending nodes) and  $\sim$ 13:30 (ascending nodes). This is important to bear in mind when considering GWs from tropical convective sources which often have a diurnal cycle (Hendon and Woodberry, 1993).

AIRS is a nadir-viewing hyperspectral sounder with 2378 infrared channels and 4 visible/near-infrared channels. This provides 3-D data on air and surface temperature, water vapour and cloud properties. AIRS observes the earth in a 90-pixel wide ( $\sim$ 1800 km) continuous swath. Its measurements have a resolution of 13.5 km (across-track) x 18 km (along-track) at the nadir, which decreases towards the edge of the swath.

We use the 3-D atmospheric temperature retrieval of Hoffmann and Alexander (2009); this retrieval covers a height range of 20 to 60 km and is specifically designed for observing stratospheric GWs. This retrieval only uses channels where the radiance originates entirely in the stratosphere and this means that the horizontal resolution does not need to be reduced to allow for cloud removal (unlike the Level 2 operational temperature retrieval - Goldberg et al. (2003)). As a result, the instrument's full sampling resolution can be leveraged. The retrieval noise is noted to be between 1.4 to 2.1 K and the vertical resolution between 7 and 13 km depending on the height (Hoffmann and Alexander, 2009).

As mentioned previously, AIRS cannot observe the full spectrum of GWs (Alexander et al., 2010; Wright et al., 2016b; Hindley et al., 2019; Hoffmann et al., 2014). Being a nadir sounder, AIRS is most sensitive to GWs with short horizontal ( $< 600 \,\mathrm{km}$ ) and long vertical wavelengths ( $> 30 \,\mathrm{km}$ ). This sensitivity decreases with increasing horizontal and decreasing vertical wavelengths. The sensitivity of AIRS was specifically characterized in Hindley et al. (2019) (Figure 2c).

# 2.2 High-resolution models

The high-resolution models come from the DYAMOND Initiative (Stevens et al., 2019b), which is a framework for high-resolution model comparison, specifically for global storm-resolving models. Here, we use data from the DYAMOND-winter project. The models in the inter-comparison all begin on the 20th January 2020, using initial and boundary data, daily seasurface temperatures and global meteorological analysis from the European Centre for Medium-Range Weather Forecasts (ECMWF) operational analysis at 9 km resolution. These models are free-running for 40 days and do not incorporate a parametrized representation of atmospheric deep convection.

All runs resolve the stratosphere and simulate the scales of waves that are visible in AIRS observations. The concept of resolution differs between observations and models. In the observations, the limitation is the sampled resolution. Theoretically, under Nyquist's theorem the sampling rate needs to be at least twice the wave frequency in order to sample the wave accurately e.g. at least two datapoints are needed to sample each wave. For AIRS observations, with a spacing (at nadir) of 13.5 km (across track) at 18 km (along track) means that waves with horizontal wavelengths >~27 km can be observed. However, in models the situation is slightly more complex, whilst the grid spacing sets the theoretical smallest resolvable scale, the actual scale at which waves can be realistically simulated depends on the model numerics and physics. As a result, 6-10 grid points are typically needed to simulate a wave in a model - this is known as the effective resolution of the model (Skamarock, 2004). In agreement with this, Preusse et al. (2014) shows that models are capable of representing waves around 8 times the resolution. As a result, the models used in our study, with horizontal resolutions between 3-5 km are able to simulate GWs with horizontal

| Dataset | Model grid | Horizontal resolution (km) | Approximate minimum visible wave size |
|---------|------------|----------------------------|---------------------------------------|
|         |            |                            | (horizontal - measured in km)         |
| AIRS    | n/a        | 13.5                       | ~27                                   |
| GEOS    | c3072      | 3                          | ~18-30                                |
| ICON    | R2B9       | 5                          | ~30-50                                |
| IFS     | TCo2559    | 4                          | ~24-40                                |

**Table 1.** Summary of observational and model datasets used.

wavelengths greater than  $\sim$ 18-50 km (see Table 1). Hence, all these models should be capable of representing waves at the horizontal scales visible to those found in AIRS.

We use the output three models from the DYAMOND-winter project, specifically: the IFS (4 km horizontal resolution), ICON (5 km resolution) and GEOS (3 km resolution). Figure 1 shows the vertical resolution of each high-resolution DYAMOND model run (ICON, GEOS and IFS) for altitudes between 15-40 km. As altitude increases, vertical resolution decreases. In this study data at 33 km altitude are used because at this height, the AIRS observations have the highest signal-to-noise ratio. At this level the GEOS model has the finest vertical resolution of approximately  $0.6 \, \text{km}$ . The IFS model has a vertical resolution of  $\sim 0.8 \, \text{km}$ . The ICON model is the most coarse, with a resolution of almost  $1.5 \, \text{km}$ . Table 1 describes the horizontal resolution of the datasets and specific details of the individual models are provided below. For a full comprehensive overview of model set up refer to Stevens et al. (2019a).

# 2.2.1 IFS

125

The ECMWF Integrated Forecasting System (IFS) is a hydrostatic model. It employs a semi-implicit, semi-Lagrangian formulation and is horizontally discretized using a spherical harmonic spectral expansion, combined with a cubic-octahedral reduced Gaussian grid (Malardel and Wedi, 2016). The DYAMOND winter simulations are run at TCo2559 horizontal resolution, corresponding to a spectral truncation at total wavenumber 2559 and an average grid spacing of 4.4 km. Time integration is performed with a time step of Δt=240 s. Vertically, the IFS uses a pressure-based hybrid η-coordinate with 137 model levels extending from the surface up to 0.01 hPa (80 km). The vertical discretization is implemented using a third-order finite element method. The IFS has a weak sponge starting at 10 hPa with a stronger sponge that comes into effect above 1 hPa. A model sponge is a damping layer that is commonly used in models to prevent instabilities resulting from the reflection of vertically propagating waves at the model top.

The DYAMOND simulation uses the Lott and Miller (1997) orographic GW drag parametrization and the non-orographic parametrization of Orr et al. (2010). These parametrizations are designed to decrease with increasing resolution and are zero at a resolution of 1.4 km (IFS Documentation). This means that at 4-km scale there is still some GW drag parametrization acting within the model. The subgrid variance in topography, which is the source for parametrized orographic GWs, scales naturally with resolution (this is the case for ICON and GEOS also). The model output is saved 3 hourly.

# 2.2.2 ICON

ICON has been developed by a German collaboration between the Deutscher Wetterdienst (DWD), the Max-Planck-Institute for Meteorology (MPI-M), the Deutsches Klimarechenzentrum (DKRZ) and the Karlsruhe Institute of Technology (KIT) and is described in Zängl et al. (2015). As the name suggests, the model is named ICON (Icosahedral Nonhydrostatic) because of its Icosahedral grid and the model is nonhydrostatic. This icosahedral grid shape allows for nearly homogeneous coverage of the earth and overcomes the problems of simpler latitude-longitude grids at the poles where longitude lines converge. Similar to the
 IFS model, ICON uses the Lott and Miller (1997) orographic wave parametrization and the Orr et al. (2010) non-orographic GW drag scheme. The DYAMOND-winter configuration is a coupled ocean-atmosphere model with a 5 km horizontal grid spacing. In the vertical there are 90 levels from the surface to 75 km, the sponge layer begins at 44 km altitude and hence model output is truncated to 44 km. The model has a timestep of Δt=4.5s, atmospheric wind and temperature output is saved 3 hourly.

## 2.2.3 **GEOS**

The Goddard Earth Observing System (GEOS) model is a flexible model that follows the modular architecture of the Earth System Modeling Framework (ESMF), which means that GEOS can be used in different configurations to support experiments. This uses finite volume (FV) dynamics on a cubed-sphere horizontal grid (Putman and Lin, 2007). The DYAMOND winter run was started on the 15th of January and 6-hourly incremental analysis updates were used to nudge the GEOS model forward from the 15th to the 20th to provide a spin up state for GEOS before releasing the free run on the 20th. The orographic GW parametrization is based on McFarlane (1987) and is scale aware. The non-orographic GW parametrization is based on Garcia and Boville (1994). The model top is at ~80 km and there are 181 vertical levels. The sponge begins at 72 km altitude. In the DYAMOND-winter version, the 3 km coupled ocean-atmosphere model is used. The model timestep is Δt=3.75s and model output is saved hourly.

# 2.3 ERA5 reanalysis

ERA5 is the fifth generation of atmospheric reanalysis from the ECMWF (Hersbach et al., 2020). ERA5 is produced by the Climate Change Service at ECMWF and provides hourly estimates of variables on a 31 km horizontal grid and resolves the atmosphere from the surface to 0.01 hPa (~80 km) altitude on 137 levels. In this work, we use background winds from ERA5 at 33 km altitude alongside the winds from each model to provide context to our results.

# 3 Method

# 170 3.1 Model sampling method

We resample each model atmosphere as though viewed by the AIRS satellite instrument to create synthetic satellite measurements which can be directly compared to AIRS observations. This is done using the sampling method developed by Wright and Hindley (2018). In summary, this method first uses the geolocation parameters of real AIRS measurements to create track



Figure 1. Vertical resolutions of the DYAMOND simulations, GEOS (orange), ICON (blue) and IFS (purple) models for altitudes 15-40 km.

files which detail the location and times of observations. To sample model output as AIRS we first use a nearest-neighbour interpolation of the model files in time. Nearest-neighbour is used instead of linear interpolation with linear interpolation you are potentially averaging to the midpoint of a wave cycle hence damping wave features.

Next, AIRS sensitivity functions are applied to account for the instrument's vertical sensitivity and inherent smoothing characteristics. These functions are based on retrieval diagnostics from the full-resolution stratospheric temperature retrieval developed by Hoffmann and Alexander (2009). In particular, the averaging kernels derived from these retrievals quantify how the retrieved temperature profiles respond to actual atmospheric temperatures and the a priori data, providing information on the vertical resolution and smoothing behaviour of the retrievals. By analyzing these diagnostics across a range of atmospheric conditions, the vertical sensitivity was approximated using a Gaussian function, with the full width at half maximum (FWHM) varying as a function of month, latitude, and whether the retrieval was performed with day- or nighttime configuration. This approach allows for a flexible and realistic emulation of the AIRS vertical sensitivity when applied to model data. An illustration of the vertical sensitivity functions used in this study is provided in the supporting information.

The horizontal sensitivity of AIRS is primarily governed by the instrument's design and optical properties. The AIRS field-of-view (FOV) spans approximately 1.1°, which corresponds to a ground footprint of about 13.5 km in diameter at nadir from its mean orbital altitude of 705 km. This resolution defines the spatial scale over which the instrument integrates radiance measurements, introducing horizontal smoothing that blends atmospheric features within each footprint. Consequently, the AIRS retrievals reflect a spatially averaged atmospheric state, limiting sensitivity to fine-scale horizontal structures such as small-scale GWs or sharp gradients. In this study, we represented the horizontal smoothing by a Gaussian function set to ap-







proximate the instrument's spatial averaging and to facilitate consistent sampling of the model output. The horizontal sensitivity representation is shown in the supporting information.

Next, the model grid is over-interpolated onto a fine grid at a resolution significantly higher than this instrument resolution. In this study,  $0.5 \, \text{km} \times 0.5 \, \text{km} \times 1/20$  of a decade of pressure (i.e., 1/20 of an order of magnitude in base 10). Finally, using this information, a weighted sum of all model fine-grid points that lie within the instrument's measurement volume is computed, which represents the model temperature corresponding to that AIRS measurement. The fine grid-point method is used to ensure that information from model grid points near to the boundaries of the sensitivity volume of AIRS are also included, not just singular points inside this volume. More details on this method can be found in Wright and Hindley (2018), which includes a detailed assessment of the sensitivity of the final results to the choice of 3D fine grid spacing. Lear et al. (2024), also applied this method in their study with a 1.4 km run of the IFS model. They compared the sampling method of Wright and Hindley (2018) (which we employ here) with a simpler sampling approach. This simpler approach, described by Hindley et al. (2021), overinterpolates the model data in the horizontal before applying Gaussian smoothing which matches the horizontal resolution of AIRS, the same process is repeated in the vertical and then the smoothed model data is evaluated at measurement locations. This simpler approach is far more computationally efficient and Lear et al. (2024) note that broadly, the results found are similar. However, there are visible differences, and it is for this reason that we use the more methodologically accurate (albeit more computationally demanding) method of Wright and Hindley (2018).

This method creates synthetic AIRS measurements of each model sampled 'as' the AIRS instrument. These datasets provide temperature data at all the sample points of AIRS and, as such, take the exact same form as AIRS data. The GWs properties are derived identically in the model-as-AIRS and the real AIRS observations. We describe this process in the following section.

# 3.2 Derivation of GW properties

We calculate GW properties from the 3D AIRS temperature retrieval of Hoffmann and Alexander (2009) and the models sampled 'as-AIRS' using a consistent method. We briefly describe this method here, but note that more detailed documentation can be found in Wright et al. (2021).

In the operational AIRS data processing, the data is split into 6 minute sections of data, known as granules. We process these granules independently. Temperature perturbations are calculated by subtracting a fourth order polynomial in the across-track direction (e.g. Alexander and Barnet, 2007; Hoffmann et al., 2014; Wright et al., 2017; Hindley et al., 2019). Figure 2 shows the temperature perturbations at 33 km altitude (in Kelvin) for the 21st January for the AIRS satellite observations and for each model sampled as though viewed by the satellite (2nd day of the model run). We see that the general morphology of GW activity matches well with activity found over Scandinavia, Northern Europe, Lake Baikal, and across the SH tropical region. However, there is more GW activity in all of the models than in the observations.

Next, the measurements are re-gridded onto a regular-in-space grid. We use a 3D spectral analysis technique colloquially known as the 2D+1 method (Wright et al., 2021). This method builds upon previous methods (Alexander and Barnet, 2007; Hindley et al., 2016; Alexander et al., 2009; Wright et al., 2017) to derive and locate GW properties of the dominant wave in three dimensions. To contextualise this, a selection of historic momentum flux derivation techniques from the AIRS satellite

**Figure 2.** Global temperature perturbations (measured in K) for the 21st January 2020. For a) AIRS observations, b) GEOS sampled 'as' the AIRS instrument, c) ICON-as-AIRS and d) IFS-as-AIRS. Data is shown at 33 km altitude.

**Figure 3.** AIRS observations of an example wave from the 6th May 2008. a) temperature perturbations at 41 km altitude, b) 3-D plot of temperature perturbations - note the exponential increase in wave amplitude with altitude has been removed in order to highlight the structure of the wave. Panels c)-e) show calculated GW momentum fluxes, specifically; c) a 2-D S-transform method similar to Alexander et al. (2009) using an S-Transform in the across track direction and a phase shift in the along track, d) AIRS observations with a collocated pass of the MLS instrument (Microwave Limb Sounder) - onboard the Aura satellite (Wright et al., 2016a) and e) the 3-D S-transform method (Wright et al., 2017)

is shown in Figure 3. The 2D+1 method that we use in this study consists of a two-dimensional Stockwell Transform in the horizontal and a phase shift method in the vertical to determine and spatially locate wave properties in all three directions. Specifically, this method is able to determine zonal, meridional, vertical wavenumbers and wave temperature amplitude for each pixel of observations. The AIRS satellite makes near-instantaneous (in time) measurements, meaning we have an ambiguity in the direction of propagation of the waves which we break by assuming that all observed waves propagate upward (Hindley et al., 2020). We calculate the absolute vertical flux of horizontal pseudo-momentum (commonly referred to as simply 'absolute MF') by,

$$\left(\mathrm{MF_{x}},\mathrm{MF_{y}}\right) = -\frac{\rho}{2m} \left[\frac{g}{N}\right]^{2} \left[\frac{T'}{\bar{T}}\right]^{2} \left(k,l\right),\tag{1}$$

with magnitude,

$$|MF| = \sqrt{MF_x^2 + MF_y^2},$$
 (2)

from Ern et al. (2004). Here, k, l, and m are the wavenumbers in the zonal, meridional and vertical directions respectively. g is acceleration due to gravity, N is the Brunt Väisälä frequency (assumed to be  $0.02\,\mathrm{s}^{-1}$ ). T' and  $\bar{T}$  are the wave amplitude and background temperature and  $\rho$  is the atmospheric density.  $\mathrm{MF_x}$  and  $\mathrm{MF_y}$  are the zonal and meridional components of momentum flux respectively.

# 240 3.3 Detecting GWs



In order to properly compare GW properties across the models and observations, we must separate the GWs from regions of no waves. Here, we use the recently developed machine learning (ML) method of Okui et al. (2025). This method uses a convolutional neural network (CNN) trained on a dataset of GWs that have been detected by the Neighbourhood Method of Berthelemy et al. (2025). We briefly describe these methods here but refer the reader to the respective original studies for further details.

Firstly, the Neighbourhood Method (Berthelemy et al., 2025) is a method that can be applied to AIRS data to identify GWs from the noisy regions of no waves using the assumption that horizontal wavelengths remain constant across the spatial extent of a wave. This means that within a wave, neighbouring pixels will have similar values of zonal and meridional wavenumber. However, in areas of noise (no wave), neighboring horizontal wavenumbers are random and bear no relationship to each other. This occurs because noise in AIRS is random, speckled noise. In their work, Berthelemy et al. (2025), used a three-dimensional S-transform method to determine wavenumbers for every pixel of data, they then applied their Neighbourhood Method to successfully identify GW events in 5 years of AIRS data.

The subsequent method of Okui et al. (2025) uses the 5-year dataset of Berthelemy et al. (2025) as a training dataset for a CNN model. Crucially, the key advancement of Okui et al. (2025) is that this machine learning algorithm can be applied to any atmospheric model output also, not just AIRS satellite observations. This is unlike Berthelemy et al. (2025) - which relies on the presence of noise in the measurements to separate wave regions from no-wave regions. An additional benefit of the Okui







et al. (2025) method is that it is applied to the temperature perturbations, not the derived GW properties, so that GW events can be identified with reduced computational demands if spectral properties are not required. This method gives a value between 0 and 1 which describes the likelihood of that pixel being part of a wave. For this work we use a threshold of 0.5 for identifying wave pixels. All pixels which are identified as a wave are included in the subsequent data analysis. Calculating properties on a pixel-by-pixel basis is consistent with previous work Hindley et al. (2020); Wright et al. (2017) and means that multiple pixels make up a single wave. As a result, the analysis will be slightly biased towards properties of waves with larger spatial areas. An alternative to this choice would be to use the CNN mask to identify individual wave packets. You could then use the average of the enclosed pixels to determine a single representative value (for each property) for the wave packet. This approach would improve the existing bias towards larger area waves, but also raises questions on how to automate the determination of wave packets and deciding on a representative value. It would also lead to an overall noisier distribution since there are fewer samples. Nonetheless, this could be an interesting technique to explore in future studies.

We apply the method of Okui et al. (2025) to temperature perturbations from AIRS, and all three model-sampled datasets at the 33 km altitude level. The same CNN model and tuning is applied for consistency.

#### 4 Results

# 4.1 Examples

Figure 4 demonstrates our methodology applied to an example overpass from the 21st January 2020. The 21st is the second day of the model runs and was chosen for these examples to avoid spurious waves from initialization shock, but before divergence of the background winds. This example shows a presumed-orographic wave event over Iceland. In the temperature perturbations we see that the wave has good similarities in magnitude and orientation in the observations and all three models, although differences can be seen especially in the ICON model. Secondly, the spatial extent of the wave (unhatched region) and location is broadly similar across all four datasets.

For the GW properties derived from the S-transform method, we see that the wave amplitude generally agrees well across AIRS, GEOS and IFS. However, it is clear that the wave represented in ICON differs. In the histograms for this wave event (panel c) we see that the models all peak at lower amplitudes (around 0.5-1.5 K) compared to the observations. Considering the horizontal wavelengths (Figure 4d1-4), the measured horizontal wavelengths generally agree across observations and models with wavelengths around ~200-300 km across the highest amplitude section. The wavelengths measured in the ICON model are shorter and this is also seen in the temperature perturbations themselves. These horizontal wavelengths match well with independent flight campaign observations of orographic waves over Iceland almost exactly four years earlier on the 25th January 2016 (Krisch et al., 2017). The Krisch et al. (2017) study used data from the infrared limb imager GLORIA onboard the German high-altitude and long-range research aircraft (HALO) which measured the waves at 10-15 km altitude. The measured wavelengths are also in agreement with the calculated horizontal wavelengths of ~250-320 km, measured by the AIRS instrument over the mountainous island of South Georgia in the southern hemisphere (SH) (Alexander et al., 2009).

**Figure 4.** Example AIRS overpass from 21st January 2020 (second day of model runs) at 33 km over the Scandinavia. Panel (a) shows the location of the overpass on the map, dashed black lines indicate the granule edge. Panel (b1-4) shows the temperature perturbations (Tp) measured in K, (c1-4) the amplitude in K, (d1-4) horizontal wavelength in km and (e1-4) vertical wavelength in km respectively. 1) Denotes AIRS observations, 2) GEOS sampled as AIRS, 3) ICON sampled as AIRS and 4) IFS sampled as AIRS. The black dashed border outlines the wave region and hatching covers regions not identified as a wave. Panels (c), (d) and (e) show histograms for the unhatched region for amplitude, horizontal and vertical wavelength respectively.

Figure 5. Same as Figure 4 but for an overpass over Madagascar on the 21st January 2020. Panel (a) shows the location of the overpass on the map, dashed black lines indicate the granule edge. Panel (b1-4) shows the temperature perturbations (Tp) measured in K, (c1-4) the amplitude in K, (d1-4) horizontal wavelength in km and (e1-4) vertical wavelength in km respectively. 1) Denotes AIRS observations, 2) GEOS sampled as AIRS, 3) ICON sampled as AIRS and 4) IFS sampled as AIRS. The black dashed border outlines the wave region and hatching covers regions not identified as a wave. Panels (c), (d) and (e) show histograms for the unhatched region for amplitude, horizontal and vertical wavelength respectively.







Outside the main wave signal, the AIRS observational data has low horizontal wavelengths and this is not the case in the models. This is because of the noise in the AIRS observations that is not present in the models. The noise in AIRS is generally pixel-to-pixel salt and pepper noise, which manifests as short horizontal wavelengths. In the models however, there is no such noise and the temperature fields are generally smooth – leading to the detection of longer horizontal wavelengths. In the vertical wavelengths (Figure 4e1-4), in the region of the wave, the vertical wavelengths in the datasets generally agree, with longer vertical wavelengths measured in the southern edge of the wave packet.

Figure 5 shows a second example from the same day of data. This example shows a far weaker wave event over Madagascar (note the smaller amplitude colourbar ranges). This is likely a convectively generated wave due to its concentric shape and lack of nearby orography. GWs from convective sources are generally more difficult to represent in models because the source is more variable compared to stationary mountain waves. Nonetheless, this wave event also matches well across AIRS observations and the model results. The curved phase fronts line up well with each other. The amplitude of the waves is broadly well matched across the datasets - except for the ICON model. However, note that this example does not intend to show that one particular model is 'better': across numerous examples, there are events that are captured in some or all or none of the model datasets. This example serves to show that the wave detection method is capable at detecting these low amplitude wave events that might otherwise be ignored if a traditional amplitude cut off approach is applied. Again we see that for the horizontal wavelengths, outside the wave region in AIRS observations, the horizontal wavelengths are around ~50 km. From this point in the paper onward, we consider time-averaged properties of wave occurrence, amplitudes and fluxes to quantify how these wave properties are represented across the models and observations.

# 4.2 GW occurrence rate

We applied our GW detection and GW property derivation method to the full 40-days of each model dataset (sampled as AIRS) and the AIRS observations to explore the spatial characteristics of GWs across the full dataset spanning 20th January - 28th February 2020. We show this analysis in ten day sections in order to capture the divergence of the models from each other and from reality.

Figure 6 shows the GW occurrence rate across 10-day periods starting from the 20th January 2020. Here GW occurrence is defined simply as percentage of times an area has a wave detected by the CNN algorithm compared to the number of times that area is observed by the satellite. Spatially, the observations and all models have GW activity across the northern hemisphere (NH) and three peaks over the eastern edge of South America, Africa and the Maritime continent between 30°S and 10°N. Outside of these two regions, GW occurrence rate is near zero in the observations and all models.

In general, across all time periods, GW occurrence rate is far lower in AIRS than it is in the models. For example, at low latitudes, AIRS observes GWs between 0-30% of the time, whereas, the IFS simulates waves the most with waves almost 90% of the time. We find that in the AIRS observations 10% of waves we identify have amplitudes lower than 1 K, whereas in the models 61%, 64% and 73% (respectively for GEOS, ICON and IFS) of waves have amplitudes lower than 1 K. This shows that the models have far more waves specifically at these low amplitudes. Figure 7 shows the same GW occurrence frequency but also excludes all waves lower than 1 K. We see that the magnitudes of GW occurrence rate are now far more comparable






between the AIRS observations and all the models, suggesting that it is in fact these low amplitude waves driving the high occurrence rates in the models in Figure 6. These low amplitude waves detected in the models could be present in the real atmosphere but it is likely that they are hard to identify over the noise in the AIRS observations explaining the differences in magnitude found in Figure 6.

In both Figure 6 and Figure 7 the NH high latitude activity aligns with the winter stratospheric polar jet with superimposed peaks over regions of orography. This is likely due to the strong, persistent eastwards winds of the jet allowing the propagation of GWs into the stratosphere and the convergence of waves into the jet. We also have the significant effect of refraction of these waves to longer vertical wavelengths which is increasing their visibility to AIRS (Noble et al., 2024; Hindley et al., 2020, 2015) which would bias the visibility in both the AIRS observations and the models sampled 'as AIRS'.

The peaks at lower latitudes correspond with heavy rainfall over Brazil, Madagascar and East Africa, and the Maritime Continent (Adler et al., 2017). The wave activity seen over the Maritime Continent and south-east from there is likely related to the South Pacific Convergence Zone. In both GEOS and IFS models, (and in both Figure 6 and Figure 7) the SH tropical GW activity is higher than ICON and the observations. This is especially the case in the IFS where off the east coast of Madagascar, GW activity is found approximately 90% of the time (with only the CNN wave detection method - Figure 6). In Corcos et al. (2021), results from Strateole 2 balloon observations show that there is a consistent background GW momentum flux of 3 mPa suggesting that GW activity in this region can be very common. On the other hand, this higher-than-expected GW occurrence of convectively generated waves in the IFS may be due to deep convection being turned off. Polichtchouk et al. (2022) compared resolved GWs in the tropical stratosphere finding that without deep convection IFS has twice as many waves as with a deep convection parametrization. They found that this was due to parametrized deep convection inhibiting convective GWs. We would also expect this to be the case with ICON (Stephan et al., 2019), however, this is not the case in our results. In addition, biases between the models and observations could arise from differences in the represented wavelengths and hence visibility within the AIRS sampling. This demonstrates that whilst GWs in the NH high latitudes are represented well, there is room for improvement in modelling convective GWs generally across the models.

Hoffmann et al. (2013) focused on characterising 'local' peak events in AIRS data using channels from the  $4.3 \,\mu m$  band which has a peak weighting height of 30-40 km altitude. Their results for NDJF (2003-2011) show similarities to our results with NH high latitude orographic hotspots and the three convective SH maxima. In their results, the NH sources do not follow a clear banded maximum across the stratospheric polar vortex, but this will be due to the larger time period used in their work and the variability of the vortex between different months and years. Their values of peak event frequencies found vary between 0-8% which is lower than our frequencies in this study using both our occurrence rate methods, however, this is likely due to difference in the methods that determine GW events.

Across the first ten days the pattern of GW occurrence rate is consistent between the AIRS observations and model datasets (Figure 6 and Figure 7). In days 11-20, the models diverge from the AIRS observations and each other. By days 31-40, the pattern in GW occurrence rate between the datasets bears little relationship to each other, as expected as the models are free-running. This is consistent with the results found in Lear et al. (2024), where there was divergence from AIRS observations and the IFS model run after approximately 7 days (synoptic timescales).

**Figure 6.** GW occurrence rate (%) defined as the percentage of AIRS overpasses with GW observations determined by the CNN model. First row AIRS observations, second row GEOS sampled as AIRS, third row ICON sampled as AIRS and bottom row IFS sampled as AIRS. Columns show average 10-day time periods since model initialisation on the 20th January 2020. Data is shown at 33 km altitude.

There is also a notable weakening in the SH tropical GW activity as time passes in AIRS, GEOS and ICON. Hindley et al. (2020) found a weakening in momentum flux in this region at the same time of year from their 18-year climatology. This is a seasonal change in the background winds which will eventually reverse in late spring (Butchart, 2022).

**Figure 7.** The same as Figure 6 but with an addition 1 K amplitude threshold for wave determination. GW occurrence rate (%) defined as the percentage of AIRS overpasses with GW observations determined by the CNN model and a 1 K amplitude threshold. First row AIRS observations, second row GEOS sampled as AIRS, third row ICON sampled as AIRS and bottom row IFS sampled as AIRS. Columns show average 10-day time periods since model initialisation on the 20th January 2020. Data is shown at 33 km altitude.







# 4.3 GW amplitudes

Figure 8 shows the 10-day conditional mean GW amplitudes from the four datasets. The conditional mean means that the average amplitude is taken across instances where there is a wave. Defined local maxima are seen over known orographic GW hotspots (Hindley et al., 2020; Hoffmann et al., 2013) including the Southern tip of Greenland, Scotland, Scandinavia and the mountainous regions in Asia. Across the first 10 days, the largest amplitudes are found over Scandinavia; here amplitudes average around 3 K in the AIRS observations and GEOS, however, this is lower in the ICON and IFS models. We also find higher amplitudes in the NH high latitude hotspots than in the SH low latitude convective regions in all datasets. Again, like Figure 6 (GW occurrence rates), there is strong morphological agreement between observations and models across the first 10 days. After 10 days, this changes (likely due to divergence in the background winds).

Generally, AIRS has higher amplitude waves in the conditional mean, than the models. This is the case in the NH high latitude orographic hotspots and in the SH low latitude convective regions, suggesting that all waves are underrepresented regardless of source. This agrees with the results shown in Lear et al. (2024) when comparing the IFS model (at 1.4 km horizontal resolution) to AIRS observations over Northern Asia, their study found that generally AIRS observed waves had higher amplitudes than in the model. Kruse et al. (2022) found that models under-represented GW amplitudes when compared to AIRS observations over the Southern Andes, even at 3 km horizontal resolution. The Kruse et al. (2022) study suggests that this could be due to waves being underresolved or overdiffused. In agreement with this and our results, Gong et al. (2015) found significantly lower wave amplitudes in ECMWF data (at 16-25 km horizontal resolution) than AIRS observations.

Figure 9 shows the unconditional mean, wherein where no wave is detected it is defined as a wave with zero wave amplitude. These results show, that in the unconditional mean, wave amplitudes are of comparable amplitudes across the observations and models. This is because fewer waves are detected in the observations and the no-wave events drag the unconditional mean down to lower values. This highlights the importance of considering how mean averages are defined in model-observational comparison studies.

# 4.4 Background stratospheric winds and precipitation

Figure 10 shows the 10-day mean stratospheric zonal winds at 33 km altitude alongside precipitation from ERA5 and each model respectively. Firstly, considering the zonal winds; across days 1-10, the zonal winds agree across ERA5 and all the models. Areas of strong zonal wind at 33 km match well with regions of strong GW activity (both in GW occurrence rate – Figure 6) and amplitude (when corresponding with orographic source regions – Figure 8). This is due to (1) strong winds allowing the propagation of GWs to the stratosphere and (2) refraction of waves to longer vertical wavelengths and increased visibility to AIRS. However, independent of the observational filter effect, more waves are observed in regions of strong winds. This was investigated by Plougonven et al. (2017) who argued that in the stratosphere, this is a result of several processes, namely source distribution, propagation and filtering but that lateral propagation of waves into regions of strong winds was an important factor. To investigate further how much the differences between the modelled waves are due to differences between the models, one could use flow-relative diagnostics of waves, as in Plougonven et al. (2017). It is possible that in both models

**Figure 8.** GW amplitudes (measured in K) 10-day conditional mean (only pixels identified as waves are considered). First row AIRS observations, second row GEOS sampled as AIRS, third row ICON sampled as AIRS and bottom row IFS sampled as AIRS. Columns show average 10-day time periods since model initialisation on the 20th January 2020. Data is shown at 33 km altitude. Data is only averaged for instances where a wave is detected.

**Figure 9.** GW amplitudes (measured in K) 10-day unconditional mean (no wave equals a wave with 0 K amplitude). First row AIRS observations, second row GEOS sampled as AIRS, third row ICON sampled as AIRS and bottom row IFS sampled as AIRS. Columns show average 10-day time periods since model initialisation on the 20th January 2020. Data is shown at 33 km altitude. Data is only averaged for instances where a wave is detected.

**Figure 10.** Panels (a-d) coloured contours show zonal mean wind at 33 km altitude and the blue shading shows the precipitation for 10-day periods starting from 20th January 2020 for ERA5 (first row), GEOS (second row), ICON (third row) and IFS (fourth row). Day 1 is the model initialisation day on the 20th January 2020. Panel e shows the topography.






(IFS and ICON), at high Northern latitudes, there are very similar PDFs of wave amplitudes conditional on the background winds. Yet, as the background flows differ substantially, maps of the GW occurrences differ substantially, but these differences do not reflect differences between the models.

Throughout days 11-20, the GW intensity is much lower in the AIRS observations than both the preceding and following 10 day periods. The GW intensity in AIRS is also lower than the model results for this period. In the background winds however, the wind strength is only slightly weaker over the Atlantic and Eurasia in ERA5 for days 11-20, than the other time-periods. This could explain the reduced GW activity detected by AIRS and shows that the detectable GW activity is very sensitive to the background wind. The strong dependence of stratospheric GWs on the background wind is also demonstrated. There is a development of a strong, persistent stratospheric polar vortex in ICON (days 21-40) which corresponds with high GW occurrence rates over this region, alongside high amplitude waves. In the IFS results, the polar vortex breaks down completely. The spatial distribution of GW occurrence and amplitudes matches this wind pattern.

Precipitation across days 1-10 is generally similar between ERA5 and each of the models. SH tropical convective GWs are found in the regions where strong precipitation lines up with stronger winds. Across South America, the precipitation is slightly stronger in the GEOS and IFS models than ICON and AIRS, which may explain why GW amplitudes and occurence rates are enhanced in this region in the GEOS and IFS models. However, we find strong precipitation over Madagascar in ICON which although appears to contribute to higher GW occurence and amplitudes, is still nowhere near as strong as those in ICON and GEOS at this time and location. Across days 11-20, precipitation rates across the datasets seem to agree somewhat more than the zonal winds. We also find that precipitation matches with areas of increased SH tropical gravity wavesm confirming that convection is the likely source of these waves. For example, the IFS model develops strong precipitation to the east of Madagascar (far more east than any other model) which coincides with eastwards-shifted peak in GW occurrences (Figure 6).

In summary, this suggest that in the northern hemisphere high latitudes this is mostly due to the evolving differences in the stratospheric background winds. Conversly, in the southern hemisphere low latitude convective regions, source variability (which we assess via model precipitation) drives the differences. This demonstrates that both accurate background winds and convective sources are vital to recreating realistic GWs in models.

# 4.5 GW Momentum Flux

Figure 11(a-d) shows the zonal and meridional GW momentum flux from the first 10 days using a two-dimensional colorbar for AIRS observations and the three model-sampled datasets. Areas where no wave is detected are defined as a momentum flux of zero (unconditional mean). We do not show the conditional mean here because when considering momentum flux, we are primarily interested in the momentum deposited over a period of time, rather than the average flux of each wave (conditional mean). Presenting both zonal and meridional momentum flux on the same figure allows us to identify the direction of the momentum flux. Traditional GWMF plots with separate zonal and meridional figures are included in the supporting material.

The spatial patterns of momentum flux agree well in location with previous satellite-derived absolute momentum fluxes for this time of year from HIRDLS (High-Resolution Dynamics Limb Sounder) and SABER (Sounding of the Atmosphere using Broadband Emission Radiometry) (Geller et al., 2013; Ern et al., 2018).






In the NH, in the observations and all models, the momentum flux is predominantly westwards (blue and purple) and we find increased momentum flux across the shape of the NH polar vortex. As discussed previously, Plougonven et al. (2017) also found increased momentum fluxes in regions of high winds that they suggest is predominantly due to lateral propagation. In the models, we also see the convergence of the waves into the centre of the polar vortex and against the background winds near Iceland. This convergence of waves into the polar vortex has been studied extensively in the SH (e.g. Sato et al., 2011; Hindley et al., 2015; Wright et al., 2017; Moffat-Griffin et al., 2020; Noble et al., 2024), but is less well characterised in the NH likely due to the greater variability of the polar vortex. In the AIRS observations and models we find peaks of activity over Newfoundland, Greenland, Scandinavia and Asia, likely associated with orographically generated waves.

In the SH low latitudes, the flux is eastwards (yellow and orange) with a meridional split in the propagation direction of these waves. This is particularly clear in the IFS over South America, the GW momentum flux is eastwards but the northern half north-eastern momentum flux whilst the southern half has south-eastern momentum flux. This divergence in meridional direction is also present in GEOS and possibly in AIRS. The momentum flux in ICON is too low to appear, probably as a result of the very low wave occurrence rates in this region (Figure 6). The directional momentum fluxes agree well with the long-term averages in Hindley et al. (2020) and Ern et al. (2017) (using a different spectral analysis method) for both regions, suggesting this 10-day period is representative of this time of year.

In Figure 11e we show zonal-mean zonal MF. There are peaks centred at 15°S (corresponding to the SH tropical GW activity) and 60°N (the NH activity around the polar vortex). For the peak in the SH tropics, AIRS and ICON are both well matched with far lower fluxes than the IFS and GEOS. However, in the NH high latitude peak, the lowest fluxes are seen in AIRS observations and ICON with IFS and GEOS overestimating the fluxes (compared to AIRS) by a factor of 3-4. It is important to remember here that this comparison is carried out within the observational filter of AIRS and that there will be other wavelengths that AIRS is less sensitive to that contribute to an overall balanced flux.

In the meridional direction (Figure 11f), for the peak in NH high latitudes, the models are more similar to each other than they are to AIRS observations. The models all show positive flux but AIRS finds negative flux. The zonal average is a residual of the many positive and negative effects and the meridional fluxes are small compared to the zonal. Likely there is slightly more negative meridional flux in the zonal mean in AIRS. This could be due to the large region of southwards flux over Russia in AIRS which is speckled northwards and southwards in the models. This is unlikely due to large-scale background wind conditions as during this initial 10-day period, the winds remain fairly similar across the models and observations.

#### 4.6 GW wavenumbers

Figure 12 shows the distribution of GW horizontal and vertical wavenumbers, and amplitudes for all four datasets. In the horizontal wavenumbers (panels a-d) we see good agreement in the shape of the distribution with the majority peak of wave activity at the lower amplitudes and longer horizontal wavelengths. However, the models have a comparatively much higher peak, consistent with our results from Figures 6 and 8, where the models had higher GW occurrences but at the lower amplitude scales than the AIRS observations.

Figure 11. (a-d) Average directional momentum flux at 33 km altitude for days 1-10, for a) AIRS observations, b) GEOS sampled as AIRS, c) ICON sampled as AIRS, d) IFS sampled as AIRS, colours correspond to the two-dimensional colorbar. Traditional separate zonal and meridional momentum flux maps are supplied in the supporting information. Panels e) and f) show zonal-mean momentum flux against latitude for zonal and meridional momentum flux respectively.


**Figure 12.** Two-dimensional histograms for (a-d) horizontal wavenumber against amplitude for the first 10 days of the model runs, (e-h) vertical wavenumber against amplitude. Panels (a, e) are AIRS observed GWs, (b, f) GEOS sampled as AIRS, (c, g) ICON sampled as AIRS and (d, h) IFS sampled as AIRS. All data is shown from 33 km altitude. Note the log10 scale on the histogram counts. White contours on the AIRS panels indicate the wavenumber-amplitude properties of areas where there is no wave present (this method is described in further detail in the text). The white cross indicates the most frequent GW properties with contours at 1/2, 1/10 and 1/100th of this maximum.

In the vertical (Figure 12, panels e-h), the distribution is also broadly similar across the datasets with the peak of wave activity occurring at the lower vertical wavelengths and low amplitudes, however, there are some differences. The models spectrum is concentrated again at the low amplitudes but with short vertical wavelengths. It is known from the limb sounders (Ern and Preusse, 2012; Preusse et al., 2014) that a broad spectrum of short vertical wavelength waves exists in the atmosphere. These waves are hard to detect for AIRS because the AIRS sensitivity makes them low-amplitude events. In the models more of these waves can be detected because the models have lower noise, which explains the difference in the spectra of detected waves and the differences in the detection statistics between AIRS and the models. We also note that the distribution in the ICON model reaches much longer wavelengths than the others.

In general, whilst the general shape of the spectrum agrees across the datasets, the models in particular contain more low-amplitude long horizontal wavelength and short vertical wavelength waves than AIRS. However, these waves (by equation 1) generally have low momentum fluxes and so a lesser impact on the general circulation. These low-amplitude waves could also exist in the real atmosphere but are not captured in the AIRS observations because of instrument noise. The machine learning wave mask (Okui et al., 2025) that we employed in this work does a good job of identifying lower amplitude waves. However, especially low amplitude waves may not even be visible in the data as they would be obscured by the noise. Additionally, future

work to compare the representation of GWs in high-resolution models across the scales (not just those visible to AIRS) would be very beneficial.

The white cross and contours in panels a) and e) indicate the GW spectrum derived AIRS satellite observations from our methods when no wave is present. This shows that where no wave is present, our methods calculate low amplitudes (to be expected) and waves with approximately  $100 \, \text{km}$  horizontal wavelength and  $\sim 17-20 \, \text{km}$  vertical wavelength. We can see that our underlying distribution of identified waves shows no skew towards these contours, suggesting that the machine learning GW identification method is robust to false positives i.e. regions of no wave that would be identified as a wave. This spectrum of 'no waves' is calculated by taking the full 40 days of AIRS measurements, calculating the average variance of each granule. We then select those granules in the lowest 20th percentile and then feed these through our standard GW analysis method.

## 5 Discussion





Our results show that state-of-the-art models can broadly capture the spatial morphology of stratospheric GWs. The comparison is particularly good across the initial 10 days of the model runs, after which the background winds diverge markedly and the GW field diverges as a result. Generally, GW amplitudes are underestimated by models in the conditional mean but match well in the unconditional mean. GW momentum flux patterns agree well in the NH high latitudes but there are discrepancies in the magnitudes of SH low latitude momentum fluxes. Here we place the results in the context of previous studies and discuss the limitations of our work.

In the conditional mean, GW amplitudes are underestimated by models compared to AIRS and this agrees with the existing literature from previous model-AIRS comparisons. Lear et al. (2024) compared ERA5 reanalysis and a 1.4 km run of the IFS model to AIRS observations over Asia finding that IFS 1.4 km model GW amplitudes were over 2 times smaller than AIRS. Kruse et al. (2022) compared a wave event over Tierra del Fuego in AIRS observations to mountain-wave resolving 10-day hindcasts using four models (IFS, ICON, WRF and UM). They found that for the 10-km resolution global models and even for the regional 3-km models, modelled GW amplitudes were reduced. Gong et al. (2015) carried out a comprehensive analysis of concentric GWs in AIRS observations and an ECMWF analysis at 16 km horizontal resolution, finding that while there was good general agreement, the convectively generated concentric GWs were still under-represented in the ECMWF analysis. Finally, Grimsdell et al. (2010) carried out a case study convective event over Australia with AIRS observations and a three-dimensional, nonlinear, nonhydrostatic cloud-resolving regional model with a horizontal resolution of 2 km. They converted the model output temperature field to a radiance field at AIRS resolution for comparison purposes. Notably, they had to multiply the heating factor in the model by a factor of 3.8 to increase the amplitude of the modelled waves to inline with the AIRS observations. This demonstrates that GW amplitudes in this model set up were also underestimated compare with AIRS observations. These studies all suggest that generally, in agreement with the results from our work, model amplitudes are underestimated.

Conversly, Hindley et al. (2021) evaluated the performance of a 1.5 km regional run of the Unified Model (UM), in a case study of GWs over South Georgia (an isolated, mountainous island in the Southern Ocean) and generally found good





agreement in amplitudes. Additionally, Okui et al. (2023), wherein, JAGUAR hindcasts (Japanese Atmospheric GCM for Upper-Atmosphere Research) are compared over a 15-day period. The JAGUAR model has particularly high vertical resolution with 340 model levels from the surface to  $\sim$ 150 km altitude with model layer spacing of 300 m in the middle atmosphere. They found no such amplitude biases. This demonstrates that whilst there is a general trend towards underestimated amplitudes in models compared to AIRS observations, this is dependent on the model and possibly the methods used. For example, when we consider the unconditional mean in our work, the amplitudes match well between all models and AIRS. When considering momentum fluxes, in our work, momentum flux magnitude agrees well across the NH high latitudes between AIRS and all three model datasets with momentum fluxes reaching -1 mPa over a latitude band ranging from 55-65°N, possibly because the total GWMF is strongly determined by a few strong peak events rather than a broad floor of low-amplitude waves. However, in the SH, AIRS agrees with ICON whilst separately, IFS and GEOS agree with each other with overestimated zonal fluxes (3-4 times larger in the zonal mean) compared to AIRS and ICON. In agreement with this, Okui et al. (2023) found that GWMF agreed well except that low latitude convective GWs are underestimated in models, suggesting that convective GWs pose a challenge for JAGUAR and the three models examined in our work. In contrast to our results which found well represented NH momentum fluxes, Lear et al. (2024) found underestimated stratospheric momentum fluxes in a 1.4 km run of the IFS and ERA5 compared with AIRS over Asia. This difference is likely be due to seasonal and methodological differences. An early intercomparison of GW momentum flux in models and observations was carried out by Geller et al. (2013), the authors compared GW momentum flux globally between observations (HIRDLS satellite, SABER satellite, radiosondes and VORCORE superpressure balloons), three climate models (with parametrized GWs) and two higher resolution models with explicit GW representation, Kanto model and CAM5 at resolutions significantly coarser than our study. Geller et al. (2013) concluded that GMWF agreed fairly well with the observations, consistent with our results for the NH. However, it is important to note that their observations are all limb scanning satellites and as a result observe a completely different proportion of the GW spectrum than AIRS.

Additionally, Stephan et al. (2019) carried out a detailed inter-comparison of six global-convection permitting simulations for the DYAMOND summer model project for the period August 2016. They found that both configurations were able to reproduce satellite-derived values from the GRACILE infrared limb sounder GW climatology (Ern et al., 2018) and AIRS (Atmospheric Infra-Red Sounder) observations. Their study showed that whilst many large-scale patterns of GW activity were represented well. There were significant differences in magnitudes. Notably, they observed no consistent improvement with higher-resolution models. However, they noted that zonal-mean GWMFs were 30-50% higher in the explicit convection simulation in the summer hemisphere tropics, convection is the dominant source of GWs here, than in the parametrized convection simulation and that this matched better with the observations. The study of Stephan et al. (2019) suggests that the representation of convection in models is an important driver of the GW behaviour similar conclusions were also drawn in the Polichtchouk et al. (2022) with the IFS. These challenges with representing convection likely explain why GWs from convective sources are variable between the models.

It is worth noting the impact of the chosen methods on the results. Our work uses a recent machine learning model to detect GWs in the satellite data (Okui et al., 2025) and this will of course have a bearing on the results and conclusions drawn.






However, when comparing the amplitude-wavenumber spectrum of AIRS detected waves (Figure 12 panel a and e), and AIRS 'no wave' granules (same Figure – white crosses) there is no bias towards the 'no wave' distribution. This indicates that whilst the machine learning method misses some wave events, it produces very few false positives. Previous observational studies have used an amplitude cut off method whereby all pixels below a cut off value are considered not to be a wave. Determining a cut off value itself can be challenging and there is no agreed threshold in the literature, choosing one that is be good for the observations may not necessarily be fair for the models. The machine learning method has the advantage of being able to detect waves below typical cut off values whilst reducing inclusion of random noisy pixels but is disadvantaged in that the machine learning is a 'black box' and may not fully capture all wave events. The machine learning method favours detecting waves in the models over the observations which is likely be a result of the machine learning method finding it challenging to detect all the waves with low amplitudes, especially when concealed by noise in the observations. Despite these limitations, we believe the machine learning approach is the most appropriate choice for this work, as it allows for consistent identification of wave events across models and observations, and avoids the need to impose arbitrary amplitude thresholds that would bias the comparison.

A final discussion point in this work is in the comparison of models with AIRS observations and the observational filter of AIRS. AIRS is particularly sensitive to waves with long vertical and short horizontal wavelengths. In sampling the models as though viewed by AIRS, we are explicitly only comparing those waves visible to AIRS. Crucially, the waves that are observed by AIRS carry a significant portion of the momentum flux and so are important to compare. Nevertheless, it would be very beneficial to compare GW properties in high-resolutions models both (1) with other satellite-based observations such as limb scanning satellites to capture a different part of the GW spectrum and (2) model-model inter-comparisons across the full spectrum without resampling as though viewed by any instrument.

# 6 Summary and Conclusions

In this work, we assess the current representation of stratospheric GWs in the DYAMOND-winter simulations in three high-resolution convection-permitting models for the period 20th January - 29th February 2020. We sample the models as if observed by the AIRS instrument and compare to real AIRS observations. This method explicitly accounts for AIRS measurement characteristics and limitations. We then apply a wave identification technique (Okui et al., 2025). This method identifies both high and low amplitude wave events in all datasets and is especially effective at detecting low-amplitude waves missed by traditional threshold-based methods. We then use three-dimensional spectral analysis techniques to calculate GW properties at 33 km altitude (the altitude where AIRS retrievals are considered best suitable to detect stratospheric GWs). From these methods we calculate and compare GW occurrence rates, amplitudes, momentum fluxes and wavelengths between the real and model-sampled data.

We conclude that:





- GW occurrence rates patterns are consistent between the models and AIRS observations. However, occurrence rates are
   higher in all models (sampled as AIRS) than in the AIRS observations. When restricting to only waves >1 K, the model occurrence rates reduce to a level which is comparable with the observations.
  - In general, higher GW amplitudes (in the conditional mean) are found in the observations than any of the model datasets, consistent with previous literature. When considering the unconditional mean, the amplitudes are better matched between the observations and the models. This highlights the importance of carefully defining the mean in model-observational comparisons such as these.
  - Global directional momentum fluxes are consistent across all models and observations. We find peaks in the NH high
    latitudes over orography and around the polar vortex, and in the SH low latitude convective regions, consistent with
    earlier studies.
    - The GWMF is westwards in the NH high latitudes and there is a meridional convergence of the momentum flux into the strong winds of the stratospheric polar vortex.
    - Magnitudes of zonal-mean zonal momentum flux in the NH high latitudes agree remarkably well with all models estimating -1 mPa of zonal-mean zonal momentum flux between 55-65°N.
    - Fluxes are directed eastwards in the SH tropics, opposite to the background stratospheric winds.
    - In the SH low latitude convective regions zonal-mean momentum flux magnitudes differ between all (model and observational) datasets. The ICON model and AIRS observations agree well, however, GEOS is almost three times stronger and the IFS model over four times stronger than observations.
  - After the first 10 days, GW activity in the AIRS observations and free-running models starts to diverge from each other.
    - In the NH this is primarily due to relative changes in the background winds. A particularly clear example of this is the development of a strong, stable polar vortex in ICON which correlates with high GW activity in the NH, compared to a breakdown of the polar vortex in the IFS model and significantly lower GW activity. This demonstrates that accurate background winds are essential for realistic GW simulations and the feedback processes between GWs and winds.
    - In the SH precipitation is an important driver of GW activity. For example, the IFS model develops strong precipitation to the east of Madagascar, driving the higher GW occurrence rates in this area.
- Overall, the models represent the spatial distribution of GWs visible to the AIRS instrument well, with peaks seen over orography in the NH high latitudes and over known convective regions in the SH. However, our results show that challenges remain with models underestimating amplitudes and overestimating GW occurrence. The NH high latitudes (which is dominated by orographic waves) is generally better represented than the SH (convective wave sources) suggesting that there there is firstly major room for improvement in the modelling of convective GWs and secondly that there are continued challenges in the SH which is historically under-observed.

These results show that we are making significant progress towards a state-of-the-art in which stratospheric models produce physically-plausible resolved GWs. However, despite increased resolution and explicit convection, the problem of underestimating amplitudes is still present, and further improvements are required before we are able to fully simulate GW characteristics which truly represent the real observed distribution.

Data availability.

The DYAMOND model simulations are available at https://easy.gems.dkrz.de/DYAMOND/index.html#getting-the-data. The 3-D AIRS temperature retrieval used in this work is described in Hoffmann and Alexander (2009) and is available at Hoffmann (2021) [Dataset]. ERA5 reanalysis used is also publicly available (Copernicus Climate Change Service, 2022).

Author contributions.


This paper was led by PN. Conceptualized by PN, HO, JA, ME, NH, LHoffmann, LHolt, AN, RP, IP, CS, MC, CK and CW. Data was curated by PN, CW, CS, WP, LHolt, LHoffmann, NH. Formal Analysis and software by PN, CW, ME, NH. Funding Acquisition, JA, ME, NH, LHoffmann, LHolt, AN, RP, IP, CS, WP, CK and CW. Resources by LHoffmann, CS, HO, ME, CW. Visualization and the writing of the original draft was carried out by PN. Everyone contributed to the validation and review and editing of the work.

Competing interests.

There are no competing interests.

Acknowledgements. This research was supported by the International Space Science Institute (ISSI) in Bern, through ISSI International
Team project 567. DYAMOND data management was provided by the German Climate Computing Center (DKRZ) and supported through
the projects ESiWACE and ESiWACE2. The projects ESiWACE and ESiWACE2 have received funding from the European Union's Horizon
2020 research and innovation programme under grant agreements No 675191 and 823988. This work used resources of the Deutsches
Klimarechenzentrum (DKRZ) granted by its Scientific Steering Committee (WLA) under project IDs bk1040 and bb1153.

PN funded by NERC grants NE/V01837X/1 and NE/W003201/1. CW was funded during this work by NERC grants NE/S00985X/1, NE/V01837X/1, NE/W003201/1 and NE/Z50399X/1 and by Royal Society University Research Fellowship URF/R/221023. L. Holt was supported by NASA Grant No. 80NSSC23K1311. NH was funded by NE/X017842/1 and NE/Z50399X/1.

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
