# Peer review of "Stratospheric gravity waves in three high-resolution models and AIRS satellite observations"

_EGUsphere, 2025_

## Referee Comment (RC2)

Review of *Stratospheric gravity waves in three high-resolution models and AIRS satellite observations by* Noble et al.

General Comment:
The authors of this paper carefully compare stratospheric GWs as detected by AIRS and as simulated by three state-of-the-art high-resolution global free-running models such as ICON, IFS, and GEOS. To ensure a fair comparison they sample the modelled temperature fields as AIRS and apply an advanced machine learning algorithm uniformly across all datasets to identify GW features. Based on this identification algorithm, the authors compare observed and simulated wave properties. Among their key findings are systematically smaller amplitudes in the simulations relative to the observations, as well as underestimated zonal-mean GW momentum fluxes in the SH low latitudes. This study is of high scientific value, as it establishes a new benchmark for model–observation comparisons. The methods are robust and the results are clearly presented. I recommend publication after the authors address a few minor revisions.

A suggestion on the structure:
Section 4 and 5 are termed *Results* and *Discussion*, respectively. However, I noticed that the authors contextualize and discuss their results quite extensively already within the Results Section. Therefore, I suggest to call Section 4 *Results and Discussion* and include lines 491-513 to Section 4.3, lines 513-540 to Section 4.5, and lines 540-555 to Section 4.6. In addition, I suggest to include Section 4.7 termed for instance *Limitations on Methods.* I suggest to include lines 556-562 to this new subsection.

A few minor suggestions:

Throughout the paper, please abbreviate northern hemisphere as "NH", southern hemisphere as "SH", gravity wave momentum flux as "GWMF" once and stick to the abbreviation from then on. In the current version it's not consistent.

Line 4: it should be "horizontally".

Line 12-13: It would be a bit clearer to say "Agreement is more consistent in the northern hemisphere (where orographic waves dominate) than in the southern hemisphere (where convective waves dominate)."

Line 82: AIRS was already introduced in line 33. Please erase "(Atmospheric Infrared Sounder)".

Line 100: I guess the numbers given here for wavelengths AIRS is sensitive to are based on the 90% sensitivity contourline in Figure 2c of Hindley et al. (2019). I would suggest to soften the criterium here a bit and rather stick to the 50% line and state "Being a nadir sounder, AIRS is most sensitive to GWs with short horizontal (<1,000km) and long vertical wavelengths (>16km)." This is simply because your histograms in Figure 4e, 5e, and 12 e-h show a lot of waves with vertical wavelengths shorter than 30km!

Line 200-207: I'd suggest to erase these lines because they distract the reader. The Lear et al. (2024) study is better placed in the discussion.

Line 225-226 and Figure 3: I'd suggest to erase this line and Figures 3c and d because you're not mentioning or referencing to them in the body of text.

Line 230: I would love to see a bit more discussion on this upward-propagation assumption in the suggested extra subsection 4.7. Dörnbrack et al. (2018) have shown significant wave activity in the lower stratosphere of downward propagating GWs during a minor SSW over Europe. What would be the consequences if an actual downward propagating wave is taken as an upward propagating wave in your analysis?

Line 261: The Hindley and Wright reference should be included via \citep{}.

Line 263-267: I'd suggest to erase these lines because they distract the reader.

Line 283-288: I'm a bit (not to say very) confused by this part. You're stating that the horizontal wavelengths as measured in ICON at 33km on the 21$^{st}$ January 2020 match well with the horizontal wavelengths measured by a completely different instrument (GLORIA) at a different altitude (10-15km) four years earlier? Furthermore, measured wavelengths (I assume in ICON) agree with calculated wavelengths (I'm not sure what the difference is here between measured and calculated wavelengths…) measured by AIRS over South Georgia. Please either erase these lines or rephrase. Admittedly, I'm not sure what point you're trying to make.

Figure 4: Please add a label to panel a). The a) is missing. Also, there is a tiny leftover title at the very top (red dots and dashes) and the "N" of "70°N" is hiding just a bit behind the image. Maybe you could fix that. Also, regarding panels c, d, and e, you might want to consider to draw bars with four different colors instead of lines. The sometimes overlap and make it hard to read. Regarding panels e1-e4, I'd also suggest to extend the colorbar towards values of 50km since the histograms indicate that waves with vertical wavelengths even larger than 50km are present.

Figure 5: Please add a label to panel a). The a) is missing. Also, the "S" in "30°S" is hiding behind the image. Like I commented before, maybe you could work on the readability of panels c-e.

Line 296: I'd suggest to erase "(note the smaller amplitude colourbar ranges)".

Line 309: Is it the 28$^{th}$ of February or the 29$^{th}$?

Line 329: Please add Sato et al. (2012) and Ehard et al. (2017) as references for the oblique propagation of waves into the jet.

Line 378: It's a bit hard to understand. I suggest to write "[…] wherein *no wave is detected* is defined as a *wave with zero amplitude*".

Line 381: Please erase "mean".

Figure 10: I think the y-labels make no sense here. Please correct them.

Line 411: Please erase the -m attached to wavesm.

Line 444-446: I think you're mixing up the two peaks in the SH and the NH. The factor of 3-4 in the fluxes is valid for the SH peak and not the NH peak. Please double check.

Figure 11: Ingenious way to visualize a vector quantity in 2D. Two tiny comments. First, on the colorwheel, please push the "N" and "S" a tiny bit to the right. They are not centered. Second, the y-ticks in panel e) do not allow for unambiguous clarity whether the y-axis is linear or logarithmic.

Line 487: Please add Gupta et al. (2024) and Garny (2025).

Line 496: There is a space missing before "Gong".

Line 505: Please add Gupta et al. (2024) and Gisinger (2022).

Line 533: Please erase "(Atmospheric Infrared Sounder)". AIRS was already introduced.

Line 538: Please add a period after "behaviour".

Line 551: Please erase "be".

Line 565: Is it the 28[th] or 29[th] of February? Please check again. It is not consistent with line 309.